# Caffeic Acid Phenethyl Ester Loaded PEG–PLGA Nanoparticles Enhance Wound Healing in Diabetic Rats

**DOI:** 10.3390/antiox12010060

**Published:** 2022-12-27

**Authors:** Mohammed Z. Nasrullah

**Affiliations:** Department of Pharmacology and Toxicology, Faculty of Pharmacy, King Abdulaziz University, Jeddah 21589, Saudi Arabia; mnasrullah@kau.edu.sa

**Keywords:** CAPE, diabetic, wound healing, skin, nanoparticles

## Abstract

Delayed wound healing is a serious complication of diabetes and a main reason for foot amputation. Caffeic acid phenethyl ester (CAPE) is a main active constituent of honeybee propolis with reported appealing pharmacological activities. In the current study, CAPE was loaded onto PEG–PLGA nanoparticles and showed a particle size of 198 ± 7.3 nm and polydispersity index of 0.43 ± 0.04. An in vivo study was performed to appraise the wound-healing activity of CAPE-loaded PEG–PLGA nanoparticles (CAPE-NPs) in diabetic rats. Wound closure was significantly accelerated in rats treated with CAPE-NPs. This was confirmed via histological examinations of skin tissues that indicated expedited healing and enhanced collagen deposition. This was accompanied by observed antioxidant activity as evidenced by the prevention of lipid peroxidation and the exhaustion of superoxide dismutase (SOD) and catalase (CAT) activities. In addition, CAPE-NPs showed superior anti-inflammatory activity as compared with the regular formula of CAPE, as they prevented the expression of interleukin-6 (IL-6) as well as tumor necrosis-alpha (TNF-α). The pro-collagen actions of CAPE-NPs were highlighted by the enhanced hyroxyproline content and up-regulation of Col 1A1 mRNA expression. Furthermore, the immunohistochemial assessment of skin tissues indicated that CAPE-NPs enhance proliferation and angiogenesis, as shown by the increased expression of transforming growth factor β1 (TGF-β1) and platelet-derived growth factor subunit B (PDGF-B). In conclusion, CAPE-loaded PEG–PLGA nanoparticles possess potent healing effects in diabetic wounds. This is mediated, at least partially, by its antioxidant, anti-inflammatory, and pro-collagen as well as angiogenic activities.

## 1. Introduction

Diabetes mellitus (DM) is a common metabolic disorder in developed and developing countries due to several risk factors such as speedy urbanization, obesity, and sedentary lifestyles [1]. Unfortunately, DM poses a heavy economic, social, and health burden [2,3]. Its complications include cerebrovascular and cardiovascular diseases, as well as nephropathy and retinopathy [4]. In addition, delayed wound healing is one of the serious diabetic complications as DM is the main reason for foot amputation [5]. The wound repair process is highly complicated and involves a sequence of molecular and cellular events [6]. In diabetes, delayed healing has been attributed to the enhanced release of inflammatory mediators such as TNF-α [7], compromised collagen formation [8], the release of growth factors such as TGF-β [9], and angiogenesis [10]. The wound healing process comprises three main stages, namely, the hemostatic, inflammatory, proliferative, and remodeling stages [11]. The standard care of diabetic foot includes surgical debridement, dressing, and infection control [12]. Furthermore, it involves the use of several re-purposed drugs that help to expedite the healing process, such as statins, phenytoin, metformin, amitriptyline, and fluoxetine [13,14,15]. In addition, natural products such as herbal-based dressings and oils have been tried due to their diverse pharmacological activities and excellent safety profiles [16].

Caffeic acid phenethyl ester (CAPE) is a main active constituent of honeybee propolis, which has been used for hundreds of years in folk medicine [17]. CAPE is a polyphenol compound with a catechol skeleton [18]. It has been shown to retain potent antioxidant and anti-inflammatory activities [19]. Furthermore, it has potent antibacterial effects [20,21] and antiviral activities [22]. This is in addition to its hepatoprotective [23,24], renoprotective [25,26], neuroprotective [27] and cardioprotective effects [19,28]. In addition, CAPE has been shown to promote the wound healing of pressure ulcers via modulating NF-κB, NOS2, and NRF2 expression [29]. It expedited the healing of mechanical nasal mucosa in rats [30] and surgically created calvarial defects in the parietal bone [31]. Furthermore, CAPE accelerated epithelium development and cutaneous healing in a rat skin wound excision model. This was confirmed histologically and was associated with significant antioxidant activity [32]. However, the medical use of CAPE is hindered by its poor water solubility and limited plasma stability with subsequent rapid clearance and low bioavailability [33]. Several attempts have been made to formulate CAPE in nano-particle preparations so as to enhance its pharmacological activities [34,35,36]. Furthermore, the wound dressing of CAPE-loaded electrospun nanofibers was shown to be biocompatible and have enhanced healing activity [37]. Indeed, PEG–PLGA copolymers that are approved by the US Food and Drug Administration (FDA) have sparked interest due to their capacity to generate a variety of NPs [38,39]. PEG–PLGA copolymers in particular are self-assembled copolymers that have amphipathic activity in water. Furthermore, PEG–PLGA NPs are biocompatible with perfect biodegradability, with a core design that is appropriate for containing active compounds with limited water solubility and an exterior hydrophilic corona that serves as an interface between the external medium and the core [39,40]. Therefore, this study aimed at evaluating the ability of CAPE-loaded PEG–PLGA nanoparticles to expedite the healing of excised wounds in diabetic rats.

## 2. Materials and Methods

### 2.1. Chemicals

CAPE > 98% was purchased from Xi’An, Julong Bio-Teck Company (Xi’An, China). Streptozotocin (STZ), polyethylene glycol-co-poly lactide-co-glycolide (PEG–PLGA) with average molecular weights of 2000 and 11,500 for PEG and PLGA, respectively, and lactide:glycolide 1:1, polyvinyl alcohol (PVA), acetone, and hydroxypropyl methyl cellulose (HPMC) were acquired from Sigma-Aldrich (St. Louis, MO, USA).

### 2.2. Animals

Male Wistar rats (210–240 g) were obtained from the vivarium of the Faculty of Pharmacy. The animals were kept in an air-conditioned facility (a temperature of 22 ± 2 °C and humidity of 60–70%) with a dark/light cycle of 12 h:12 h with free access to tap water and standard food pellets. Animal handling procedures and protocol were permitted by the Research Ethics Committee (REC) of the Faculty of Pharmacy, KAU, Saudi Arabia (Reference # PH-1443-73).

### 2.3. Preparation of CAPE-NPs

The previously described technique was used to create CAPE-loaded PEG–PLGA NPs [14]. In a summary, 10% *w*/*w* CAPE and PEG–PLGA were dissolved in acetone before the organic solution was added gradually to an aqueous solution of polyvinyl alcohol (PVA) that contained 0.5% *w*/*v* PEG–PLGA. The dispersion underwent probe sonication for one minute after being agitated for six hours at room temperature to completely remove the organic solvent. The created NPs were centrifuged at 15,000 g for 30 min at 4 °C (Beckman Coulter, Fullerton, CA, USA). The separated NPs were redispersed, and any free CAPE or extra PVA was removed by centrifuging them twice in Milli-Q water. The CAPE–NPs were lyophilized using mannitol (5% *w*/*v*) as a cryoprotectant. With a few minor adjustments, CAPE-loaded hydrogel and CAPE self-assembled PEG–PLGA-loaded hydrogel were created. In brief, lyophilized CAPE self-assembled PEG–PLGA powder or its equivalent in raw CAPE was dispersed in double-distilled water. HPMC at a concentration of 1.5% was added to the aqueous dispersion. Prior to further usage, the gels were stored at 4 °C for 24 h. CAPE or its equivalent nanoformula was present in the hydrogel at a concentration of 2% *w*/*w*.

### 2.4. Characterization of CAPE-NPs’ Size

The size of the CAPE-NPs was measured using the Zetasizer (Malvern, Worcestershire, UK). Before analysis, the samples were diluted in deionized water. Measurements were performed in triplicate.

### 2.5. Evaluation of CAPE-PEG–PLGA Loading Capacity

CAPE loading was determined in the prepared CAPE-PEG–PLGA by dissolving a known excess amount of CAPE in 1 g of plain PEG–PLGA separately. The mixture was kept at 25  ±  2  °C for 24 h in a shaking water bath. This was followed by centrifugation at 4000 rpm for 15 min. The precipitates were collected, washed, and dispersed in methanol. Then, the drug was extracted, and the amount of CAPE was determined using the previously mentioned HPLC method [41]. The detection wavelength was set at 325 nm. The flow rate was 0.4 mL/min. The column temperature was 25  °C. The drug loading capacity was determined using the following equation:$$\mathrm{CAPE}\;\mathrm{loading}\;\mathrm{capacity}=\frac{\mathrm{CAPE}\;\mathrm{content}\;\mathrm{in}\;\mathrm{the}\;\mathrm{prepared}\;\mathrm{CAPE}-\mathrm{PEG}-\mathrm{PLGA}\;\left(\mathrm{mg}\right)}{\mathrm{Total}\;\mathrm{weight}\;\left(\mathrm{mg}\right)}\;\times \;100$$

### 2.6. In Vitro CAPE Release Study

The in vitro release of CAPE was performed using a dialysis bag with a molecular weight cut-off of 12,000 Da. Briefly, the CAPE-NPs were introduced into the dialysis bag, tied, and immersed in the release medium. The release medium consisted of phosphate-buffered saline (PBS) with a pH of 6.8 (500 mL) containing Tween 80 (0.5%). The system was maintained at 37 °C in a shaking water bath. The samples were withdrawn at time points of 0, 2, 4, 6, 12, 24, and 48 h and analyzed for CAPE content with the HPLC method [41].

### 2.7. Experimental Design

Hyperglycemia was induced via a single i.p. injection of streptozotocin (STZ) in a citrate buffer (100 mM, pH 4.5) at 55 mg/kg for each rat after 18 hours of fasting [42]. Fasting blood glucose (FBG) was measured daily using a glucometer for two weeks after the STZ injection, and only animals with sustained FBG higher than 250 mg/dL were included. For the chosen diabetic animals, FBG was monitored once weekly over the course of the experiment. To induce wounds, diabetic rats were i.p. injected with ketamine/xylazine at a dose of 50/5 mg/kg, respectively [43]. Subsequently, the skin of each rat at the dorsal surface was exposed by shaving and sterilized with an antiseptic iodine solution. After that, an excision circle of 1 cm diameter was made on each rat’s dorsal surface, followed by disinfection. A local anesthetic of lidocaine-HCl (2%) and 1:80,000 epinephrine was injected subcutaneously (4.4 mg/kg) adjacent to the wound area instantly after excision to lessen pain.

Wounded diabetic rats were then divided into 5 groups (10 in each): Group (1) comprised untreated control rats with no treatment; Group (2) received the topical vehicle used in the preparation of the CAPE-NPs on the wound area; Group (3) received a topical application of CAPE-raw prepared in hydrogel on the wound area (0.5 g); Group (4) received a topical application of CAPE-NPs, and Group (5) comprised positive control rats treated with the commercial ointment Mebo^®^ (Gulf Pharmaceutical Industries, Ras Al Khaimah, UAE) on the wound area. The ointment contained β-sitosterol as the main active ingredient. All treatments were topically applied (0.5 g) on a daily basis for 14 days. Wounds were covered with sterile dressings that were changed every day. Under current experimental conditions, wound diameters (WD) were measured, and the wounds were photographed on day 0, 3, 7, 10, and 14. On day 7, 4 animals/group were sacrificed by decapitation, and the skins of the wound areas were dissected out and fixed in 10% neutral formalin. On day 14, the rest of the animals in all groups were sacrificed, and the skins of the wound areas were dissected out. Part of the skin obtained from each animal was fixed in 10% formalin. The remaining part was kept at −80 °C. For homogenization, skin tissues were carefully washed with ice-cold saline and softly dried using filter papers. Then, homogenates were prepared in ice-cold phosphate-buffered saline (PBS, 100 mM, pH 7.4)

### 2.8. Assessment of Wound Contraction

The percentage of wound contraction was computed from the difference in wound diameter (WD) between days 0 and 14, according to the formula [44]
$$\mathrm{Wound}\;\mathrm{contraction}\;\%\;=\frac{\mathrm{Day}\;\left(0\right)\;\mathrm{WD}-\mathrm{Day}\;\left(14\right)\;\mathrm{WD}}{\mathrm{Day}\;\left(0\right)\;\mathrm{WD}}\times 100$$

### 2.9. Histopathology

Skin tissues collected from wounded areas on days 7 and 14 were fixed in neutral formalin (10 %) for 24 h. This was followed by the preparation of a paraffin block and sectioning at 5 µm thickness. Some sections were stained with H&E and the remaining were stained with Masson’s trichrome (MT) specific to collagen fibers [45]. Additional sections were kept unstained on positively charged slides for subsequent immunohistochemical studies.

### 2.10. Biochemical Analyses

MDA tissue concentration as well as the enzymatic activities of SOD and CAT were determined using commercially available biochemical kits, based on the manufacturer’s instructions (Biodiagnostic, Giza, Egypt). The skin content of hydroxyproline was measured with a specific ELISA kit (Cat. # MBS726071, MyBioSource, San Diego, CA, USA) according to the manufacturer’s instructions. Total protein content in skin tissue homogenates was assessed using a colorimetric BCA protein assay kit (Cat. #23227), purchased from Thermo-Fisher Scientific^®^ (Vienna, Austria).

### 2.11. Quantitative Real-Time PCR (qRT-PCR)

An ultrasonic probe was utilized to homogenize skin tissues, followed by RNA extraction with a nucleic acid extraction kit (NucleoSpin^®^, Macherey-Nagel GmbH & Co. KG, Duerin, Germany). The purity expressed as a A260/A280 ratio along with RNA concentrations were measured spectrophotometrically. Reverse transcription was accomplished using a commercially available reverse transcription kit (Applied Biosystems, Foster City, CA, USA). PCR amplification reactions were performed using a Taq PCR Master Mix Kit (Qiagen, Valencia, CA, USA) coupled to the primers shown in Table 1. Data were expressed in the cycle threshold (Ct). The PCR datasheet included the Ct values of the assessed gene (Collagen 1A1) and the housekeeping gene (GAPDH). The relative quantitation (RQ) was quantified according to the calculation of delta-delta Ct (ΔΔCt).

### 2.12. Assessment of IL-6, TNF-α, TGF-β1, and PDGFRα, Immunohistochemically

Skin sections were first blocked with 5% bovine serum albumin (in TBS) for 120 min, then incubated overnight at 4 °C with the specified primary antibodies (1:500): mouse monoclonal antibodies to TNF-α Cat. # (ab220210) and IL-6 Cat. # (ab9324), rabbit monoclonal antibody to TGF-β1 Cat. # (ab215715), and rabbit polyclonal antibody to PDGFRβ Cat.# (ab51046). All antibodies were purchased from Abcam, Cambridge, UK. Next to washing, slides were then incubated with the appropriate secondary antibodies for 90 min. Sections were then stained with 0.02% DAB reagent containing 0.01% hydrogen peroxide. A counter stain (hematoxylin) was used, and slides were inspected under a light microscope [46]. Optical density (OD) was determined using image analysis software (ImageJ, 1.48a, NIH, Bethesda, MD, USA).

### 2.13. Statistical Analysis

Data are shown as mean ± (S.D). The statistical difference among means was tested using one-way ANOVA followed by Tukey’s post hoc test. GraphPad Prism software^®^ (version 8.0, San Diego, CA, USA) was used to perform all analyses. *p* values < 0.05 were considered statistically significant.

## 3. Results

### 3.1. Particle Size, Polydispersity Index, and Loading Capacity

The CAPE-NPs were created using the nanoprecipitation technique. The CAPE-NPs had an average particle size of 198 ± 7.3 nm (Figure 1) and a polydispersity index of 0.43 ± 0.04. Furthermore, the CAPE loading capacity was found to be 81.73 ± 5.1.

### 3.2. Assessment of In Vitro CAPE Release

The in vitro release profile of CAPE from the prepared formula in a pH 6.8 phosphate buffer is shown in Figure 2. The release of CAPE was obviously sustained for 48 h. A slight burst release was observed at 2 h, amounting to 24.1 ± 1.8%. This was followed by sustained release of CAPE that reached a cumulative value of 77.8 ± 7.9% and 98.6 ± 10.7% at 24 and 48 h, respectively.

### 3.3. Effect of CAPE-NPs on Wound Contraction

As shown in Figure 3A, the untreated group and vehicle-treated group, CAPE-raw formulation, CAPE-NP formulation, and positive control formulation had open wounds on day 0. However, on days 3, 7, 10, and 14, the topical application of the CAPE-NPs exhibited an obvious and significant enhancement in wound contraction followed by the marketed formulation (positive control) and CAPE-raw formulation. It is noteworthy that the diabetic rats treated with the CAPE-NPs exhibited a wound contradiction of 97% on day 14 (Figure 3B).

### 3.4. Effect of CAPE-NPs on Histological Changes of Wounded Skin (H&E and MT Staining)

Figure 4 indicates that the epidermal layer of the untreated and vehicle-treated rat skins displayed obvious focal hyperkeratosis and acanthosis. In addition, the dermal layer showed an increased deposition of collagen fibers along with scattered inflammatory cell infiltration. However, animals treated with the CAPE-raw and CAPE-NPs showed marked reductions in histopathological as well as inflammatory changes. These cells were identified morphologically by their small size and basophilic cytoplasm as well as rounded nuclei. It was obvious that the CAPE-NP-treated animals showed the highest degree of collagen deposition, as indicated by blue coloration in MT-stained sections with scattered areas reaching advanced phases of wound healing.

### 3.5. Effect of CAPE-NPs on Antioxidant Status

Figure 5A indicates that skin specimens collected from the untreated and vehicle-treated animals showed a significantly higher content of the lipid peroxidation marker MDA. However, the CAPE-NP group showed potent activity in inhibiting MDA in comparison with all the other groups. In addition, surgical wounding in diabetic rats was accompanied by a significant reduction in SOD and CAT enzymatic activity. Nevertheless, treatment with CAPE-raw, CAPE-NPs, and the commercial positive control significantly enhanced SOD activity by 28.6, 59.5, and 39.3%, respectively, as compared with the untreated group (Figure 5B). Similarly, CAT activity was significantly enhanced by 31.3% and 20.2% for the CAPE-NP and the positive control groups, respectively, in comparison with the untreated animals (Figure 5C).

### 3.6. Impact of CAPE-NPs on Inflammatory Markers

The assessments of IL-6 and TNF-α expression indicated that the untreated group as well as the vehicle-treated group showed the highest elevations of both markers. However, the application of CAPE-Raw, CAPE-NPs, and the positive control expedited the inflammatory phase and significantly reduced the expression of its markers. The CAPE-NP and positive control groups showed the highest inhibitory potencies, as IL-6 expression was 63.7 and 71.9%, respectively, as compared with the untreated control group. Similarly, the CAPE-NP and positive control rats showed TNF-α expressions as low as 65.4% and 77.1% of the untreated animals, respectively (Figure 6).

### 3.7. Effect of CAPE-NPs on Markers of Collagen Formation

The application of the CAPE-raw or CAPE-NPs formulations significantly increased the hydroxyproline content by 51.3% and 113.6%, respectively, as compared with the untreated group (Figure 7A). Likewise, both preparations significantly up-regulated the mRNA of Col1A1 by 180% and 230%, respectively, as compared with the untreated animals (Figure 7B).

### 3.8. Impact of CAPE-NPs on Expression of TGF-β1 and PDGF-B

As depicted in Figure 8, the treatment of diabetic rats with CAP-NPs significantly enhanced TGF-β1 expression in comparison with the untreated, vehicle-treated, and CAPE-raw groups, by 110.8, 81.4, and 22.0%, respectively. In addition, PDGFR-β was significantly enhanced by the application of CAP-NPs by 115.8%, as compared with untreated animals. It is noteworthy to report that the CAPE-NP group was superior to the positive control group in enhancing PDGFR-β expression by 20.0%.

## 4. Discussion

Delayed wound healing is a main sign of diabetes and may lead to chronic infection, diabetic ulcers, or even limb amputation [47]. The successes of currently available topical medications are unsatisfactory, and the incidence of diabetic foot ulcers is still relatively high [48]. CAPE is an active constituent of honeybee propolis and possesses a plethora of appealing pharmacological activities [17]. The application of nanotechnology in the pharmaceutical sector has received more attention in recent years. New biodegradable self-assembled NPs of the PEG–PLGA amphiphilic diblock copolymer based on CAPE were prepared in the current work. The most popular nanocarriers for the administration of a range of drugs are biocompatible nanocarriers, notably biodegradable nanoparticles [49]. The assessment of the release of CAPE indicated that complete release needs more than 24 h. Thus, it can be concluded that the polymeric micelles formed on the PEG–PLGA block copolymer provided a platform that could sustain CAPE for more than 24 h. The entrapment of the hydrophobic CAPE in the PLGA core could be credited to its strong drug-sustaining behavior [50]. These data are in line with previous reports on PEG–PLGA-based formulae [51]. The aim of this study was to evaluate the ability of CAPE-loaded biodegradable PEG–PLGA NPs to enhance wound healing in diabetic rats. Our data indicated that CAPE-NPs significantly expedited the rate of wound healing in diabetic rats as compared with CAPE-raw and a commercially available preparation. This was confirmed by our histological findings that indicated an enhanced proportion of wounded skin in phase III healing with reduced inflammatory cell infiltration and increased collagen deposition. This is consistent with previous reports highlighting the ability of CAPE to accelerate cutaneous healing in normoglycemic rats [32]. The observed enhancement in healing activity with the nano-CAPE preparation is in line with the accumulating literature highlighting the advantages of using nanotechnology in topical drug delivery. These include the sustainable and controlled release of encapsulated active ingredients for a desired period until wound healing [52,53]. In addition, nanoparticles confer better interactions and penetration at the wound sites, making them excellent at tackling the complexity of diabetic wound healing [54,55].

Excessive oxidative stress and a decreased antioxidant ability of skin tissues are major contributors to non-healing diabetic wounds [56]. In this study, CAPE in both raw and nano preparations exhibited potent antioxidant activity as evidenced by its inhibition of accumulation of lipid peroxidation products and exhaustion of SOD and CAT activities. This is supported by the reported antioxidant actions of CAPE [57,58,59]. This has been attributed to the chemical structure of CAPE, which chelates transitional metals and stabilizes free radicals [60]. This is in line with the current recommendations for the use of molecules with antioxidant activity to accelerate diabetic wound healing [61]. In particular, bionanomaterials have found an important place in the management of diabetic wound healing [62].

Prolonged inflammation has been suggested as a major player in the pathogenesis of delayed wound healing [63,64]. Inflammatory mediators adversely affect growth factor expression and fibroblasts’ function [65]. In the current study, delayed wound healing in diabetic animals was associated with enhanced expressions of IL-6 and TNF-α. Fortunately, CAPE in both formulations significantly down-regulated their expressions. This is supported by several studies reporting the anti-inflammatory properties of CAPE [66,67,68]. Moreover, CAPE’s anti-inflammatory activities have been previously implicated in burn healing in rats [69]. In addition, several studies indicated that the delayed healing of diabetic wounds is associated with decreased collagen deposition independent of glycemic control [70]. In line with this suggestion, our data showed that untreated wounds had a decreased content of hydroxyproline together with a down-regulation of Col 1A1 mRNA expression. The beneficial effects of CAPE were associated with pro-collagen activities. These findings gain support by the previous study that highlighted the pro-collagen properties of CAPE as indicated by the accumulation of hydroxyproline in burn wounds [69]. This is also in harmony with previous histological studies that indicated that CAPE accelerates cutaneous wound healing via enhancing collagen deposition [32].

Growth factors play a vital role in wound healing by regulating cellular proliferation and differentiation [71]. Unfortunately, the production of several growth factors responsible for initiating and sustaining the healing process is impaired in diabetes [13]. Experimental studies showed that the application of TGF-β administered in a collagen gel increased the healing of incisional wounds [72]. Our data indicated a decreased expression of TGF-β in tissues collected from diabetic untreated wound tissues. On the other hand, the application of the CAPE formulae resulted in an enhancement in TGF-β expression. This supports the observed collagen deposition and enhanced wound healing. In addition, it has been reported that the inflammatory phase is characterized by the moving of fibroblasts and the release of a variety of angiogenic growth factors at the location of the wound by keratinocytes [73]. A key reason for many non-healing wounds is inhibited angiogenesis. Therefore, several strategies have been adopted to enhance angiogenesis in wound healing [74]. In this study, CAPE’s healing activity was associated with an increased expression of PDGFR-β. Several experimental and clinical studies have demonstrated the role of the angiogenic growth factor PDGF [75,76]. This possible role of CAPE in promoting the biosynthesis of extra-cellular matrix, epithelization, and angiogenesis has been previously described [17]. In conclusion, CAPE-loaded PEG–PLGA nanoparticles possess potent healing effects in diabetic wounds. This is mediated, at least partially, by their antioxidant, anti-inflammatory, pro-collagen, and angiogenic activities.

## 5. Conclusions

In conclusion, CAPE-loaded PEG–PLGA nanoparticles possess potent healing effects in diabetic wounds. This is mediated, at least partially, by their antioxidant, anti-inflammatory, pro-collagen, and angiogenic activities.

## Figures and Tables

**Figure 1 antioxidants-12-00060-f001:**
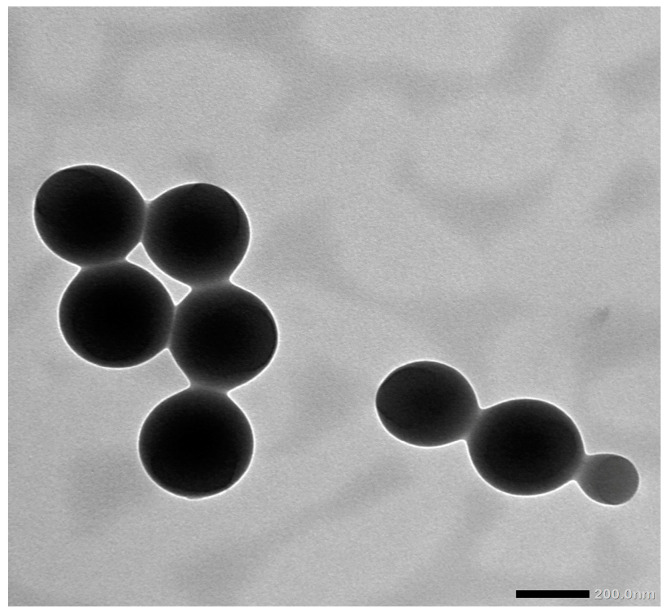
Transmission electron microscopy image of CAPE-PEG–PLGA NPs showing their particle size.

**Figure 2 antioxidants-12-00060-f002:**
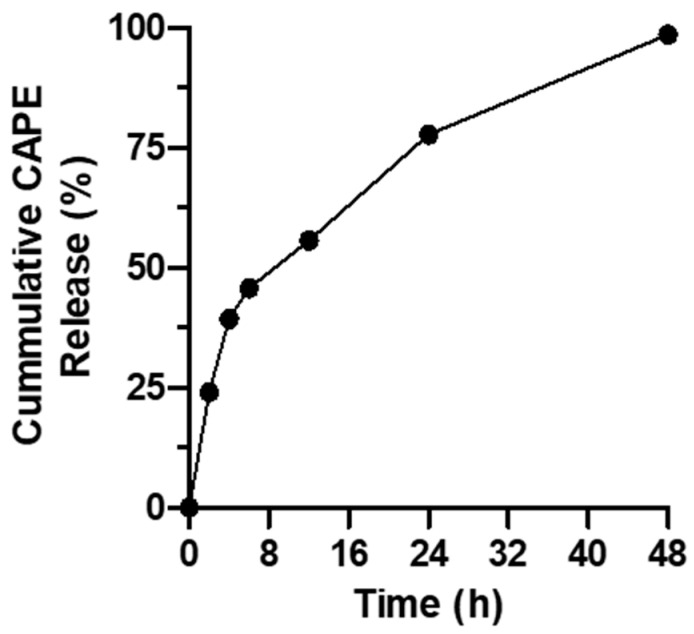
In vitro release of CAPE from CAPE-PEG–PLGA in phosphate-buffered saline (PBS) at 37 °C, pH 6.8 containing Tween 80 (0.5%) as a solubilizer.

**Figure 3 antioxidants-12-00060-f003:**
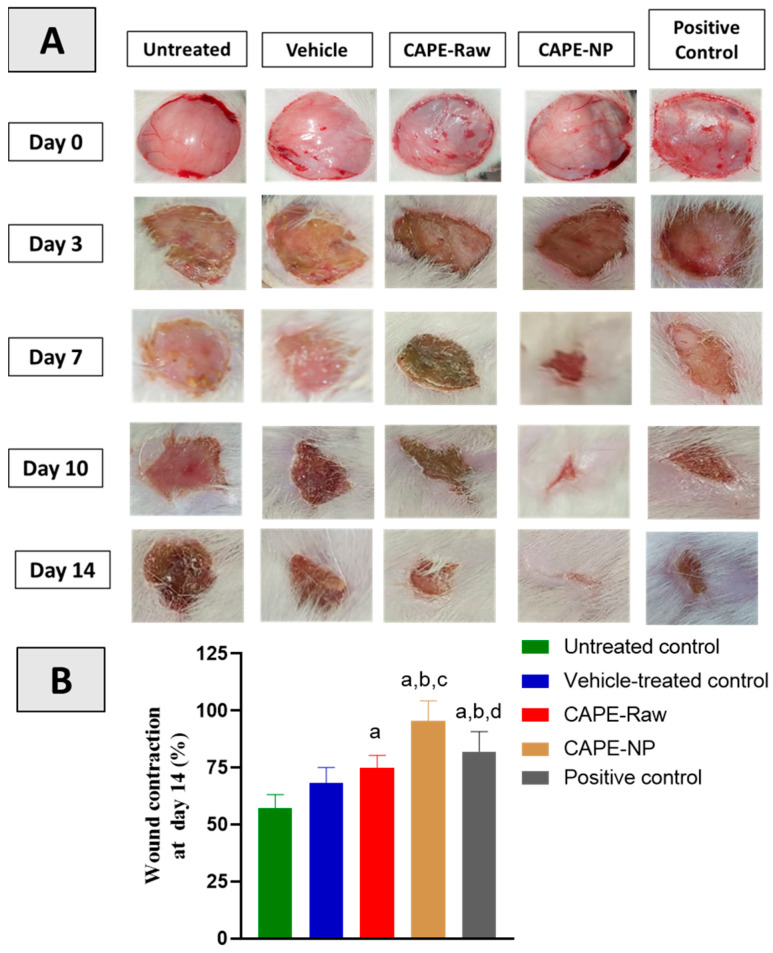
(**A**) Effects of CAPE-NPs on wound healing and % wound contraction on days 0,3,7,10, and 14. (**B**) Graphic presentation of wound contraction at day 14.Data are expressed as mean (n = 6) ± SD. (a) Significantly different from untreated control at *p*  <  0.05, (b) significantly different from vehicle-treated control at *p*  <  0.05, (c) significantly different from CAPE-raw at *p*  <  0.05, and (d) significantly different from CAPE-NP at *p*  <  0.05.

**Figure 4 antioxidants-12-00060-f004:**
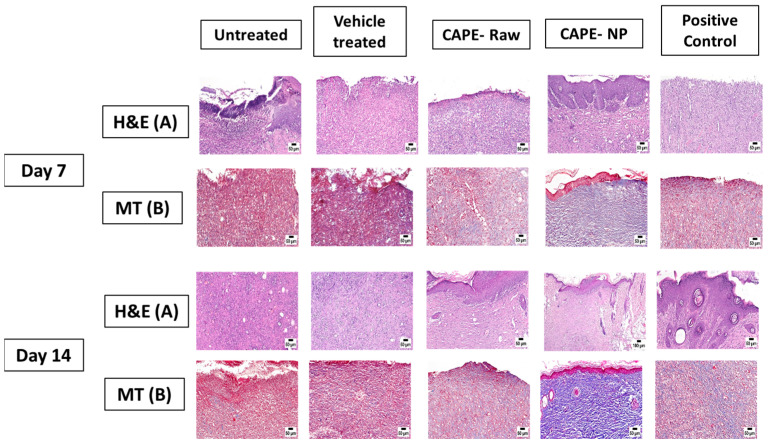
Effects of CAPE-NPs on skin histological changes. (**A**) H&E staining and (**B**) MT staining.

**Figure 5 antioxidants-12-00060-f005:**
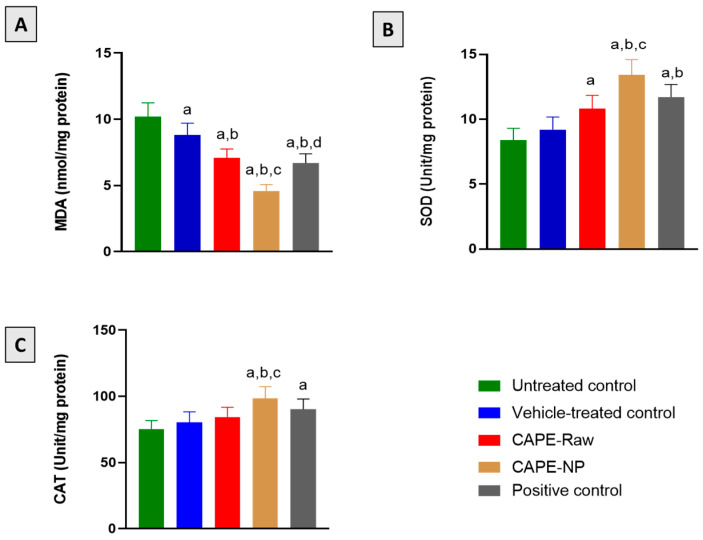
Effects of CAPE-NPs on oxidative status markers (**A**) MDA, (**B**) SOD, and (**C**) CAT. Data are expressed as mean (n = 6) ± SD. (a) Significantly different from untreated control at *p*  <  0.05, (b) significantly different from vehicle-treated control at *p*  <  0.05, (c) significantly different from CAPE-raw at *p*  < 0 .05, and (d) significantly different from CAPE-NP at *p*  <  0.05.

**Figure 6 antioxidants-12-00060-f006:**
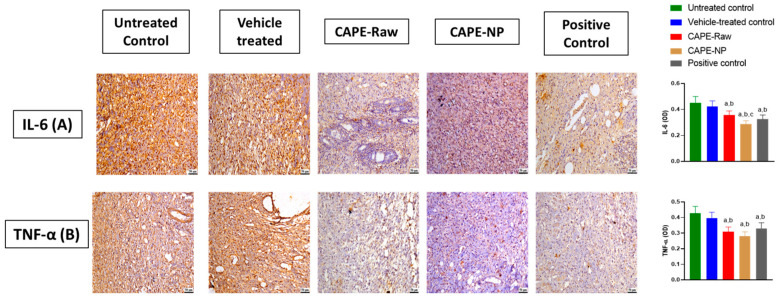
Effects of CAPE-NPs on the expression of (**A**) IL-6 and (**B**) TNF-α. Data are expressed as mean (n = 6) ± SD. (a) Significantly different from untreated control at *p*  <  0.05, (b) significantly different from vehicle-treated control at *p*  <  0.05, and (c) significantly different from CAPE-raw at *p*  < 0 .05.

**Figure 7 antioxidants-12-00060-f007:**
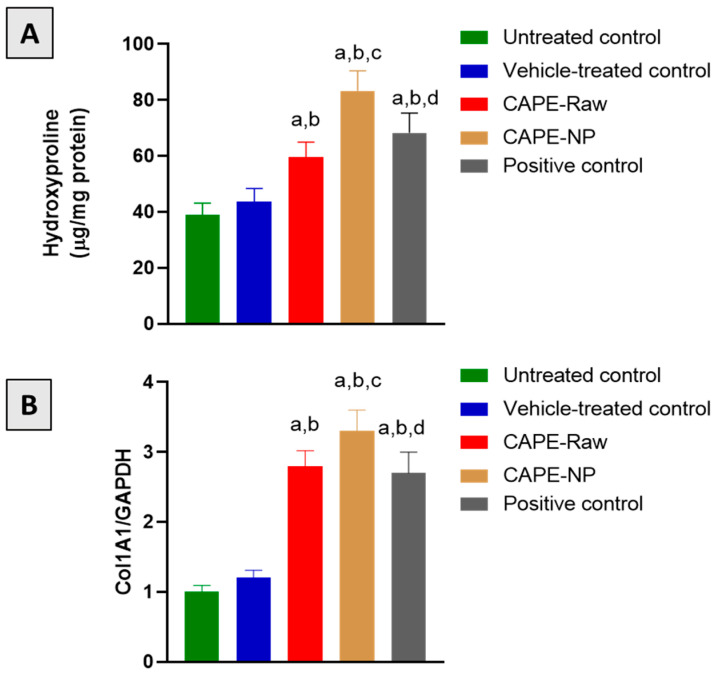
Effect of CAPE-NPs on skin content of hydroxyproline (**A**) and mRNA expression of Col 1A1 (**B**) in diabetic rat wounds on day 14. Data are expressed as mean (n = 6) ± SD. (a) Significantly different from untreated control at *p*  < 0 .05, (b) significantly different from vehicle-treated control at *p*  <  0.05, (c) significantly different from CAPE-raw at *p*  <  0.05, and (d) significantly different from CAPE-NP at *p*  < 0 .05.

**Figure 8 antioxidants-12-00060-f008:**
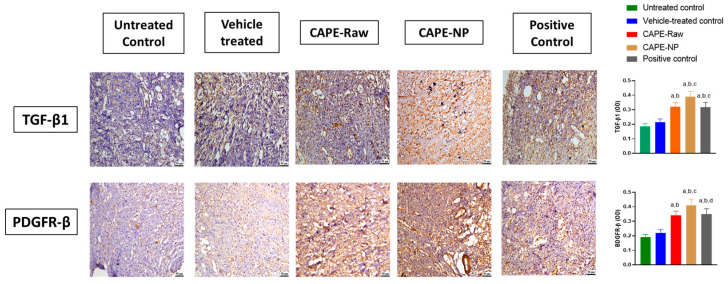
Effect of CAPE-NPs on TGF-β (upper row), and PDGF-B (lower row) expression in diabetic rat skin wounds on day 14. Data are expressed as mean (n = 6) ± SD. (a) Significantly different from untreated control at *p*  <  0.05, (b) significantly different from vehicle-treated control at *p*  < 0 .05, (c) significantly different from CAPE-raw at *p*  < 0 .05, and (d) significantly different from CAPE-NP at *p*  < 0 .05.

**Table 1 antioxidants-12-00060-t001:** The list of primers used for RT-qPCR.

	Forward	Reverse	Accession Number
Col1A1	ATCAGCCCAAACCCCAAGGAGA	CGCAGGAAGGTCAGCTGGATAG	NM_053304.1
GAPDH	CCATTCTTCCACCTTTGATGCT	TGTTGCTGTAGCCATATTCATTGT	NM_017008.4

## Data Availability

Data are available within the article or from the corresponding author upon reasonable request.

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
