# Peer review of "Caffeic Acid Phenethyl Ester Loaded PEG–PLGA Nanoparticles Enhance Wound Healing in Diabetic Rats"

_antioxidants, 2022, doi:10.3390/antiox12010060_

Round 1
Reviewer 1 Report
In the present work, PEG-PLGA nanoparticles were designed to be used for CAPE loading. CAPE is used for the treatment of diabetic complications due to its advantages in antioxidant, anti-inflammatory, and pro-collagen production, among others, which promote wound healing. Loading CAPE with PEG-PLGA nanoparticles enhances the bioavailability of CAPE and allows it to show obvious benefits in wound healing. This article is well-designed, and easy to read and the experimental results are clearly described. However, there are some spelling errors and the interpretation of the experimental results is quite simple. There are some suggestions made to combat this and hopefully it will be helpful for its improvement.
1. Line 98. Does the "2% (w/w)" in this sentence refer to the amount of CAPE or the amount of nanoparticle powder? and what is the loading capacity of CAPE into the nanoparticles? How is it calculated?
2. Line 119. What is the content of the β-sitosterol in the commercial sample used in "Group 5" in this sentence? Is it similar to the amount of CAPE in the experimental groups?
3. Line 193. Misspelling of "a wounded contradiction".
4. Line 204. How does "CAPE-NP showed marked reductions in the inflammatory changes" show up?
5. Line 282. How does "These include the sustainable and controlled release of encapsulated active ingredient" show up?
6. Figure 1. The scale of the figure is not clearly shown.
7. Figures 3 and 5. "(A)" and "(B)" are not marked in the figures.
8. Figure 7. Misspelling of " upper raw", and "VEGF-A" is not shown in the figure.
Author Response
Response to the comments of Reviewer 1:
In the present work, PEG-PLGA nanoparticles were designed to be used for CAPE loading. CAPE is used for the treatment of diabetic complications due to its advantages in antioxidant, anti-inflammatory, and pro-collagen production, among others, which promote wound healing. Loading CAPE with PEG-PLGA nanoparticles enhances the bioavailability of CAPE and allows it to show obvious benefits in wound healing. This article is well-designed, and easy to read and the experimental results are clearly described. However, there are some spelling errors and the interpretation of the experimental results is quite simple. There are some suggestions made to combat this and hopefully it will be helpful for its improvement.
- Line 98. Does the "2% (w/w)" in this sentence refer to the amount of CAPE or the amount of nanoparticle powder? and what is the loading capacity of CAPE into the nanoparticles? How is it calculated?
RE: The reviewer comment is greatly appreciated. In fact, 2% (w/w) refers to the amount of nanoparticle powder. In addition, CAPE was determined by HPLC method as described in the methodology under section 2.5. In response to the reviewer suggestion, we have made an in-vitro release study. Kindly, check section 3.2. Accordingly, Materials & Methods and Discussion were modified.
- Line 119. What is the content of the β-sitosterol in the commercial sample used in "Group 5" in this sentence? Is it similar to the amount of CAPE in the experimental groups?
RE: 0.25% and sesame oil and beeswax as base of the ointment.
- Line 193. Misspelling of "a wounded contradiction".
RE: contraction
- Line 204. How does "CAPE-NP showed marked reductions in the inflammatory changes" show up?
RE: Histological investigation indicated decreased number of inflammatory cells infiltration in CAPE-PEG-PLGA treated groups. These cells were identified morphologically by their small size and basophilic cytoplasm as well as rounded nuclei (PMID30002960)
- Line 282. How does "These include the sustainable and controlled release of encapsulated active ingredient" show up?
RE: Kindly, check the in vitro release study under section 3.2
- Figure 1. The scale of the figure is not clearly shown.
RE: the figure resolution has been enhanced (200nm) to 600 dpi
- Figures 3 and 5. "(A)" and "(B)" are not marked in the figures.
RE: Done
- Figure 7. Misspelling of " upper raw", and "VEGF-A" is not shown in the figure.
RE: row not raw, remove VEGF-A.
Reviewer 2 Report
Introduction:
(1) The authors described the meaning of the improvement of the solubility of CAPE and the encapsulation of CAPE in PEG-PLGA.
(2) The solubility of CAPE should be described in specific figures.
(3) Why did the author choose PEG-PLGA? The author added a description of the properties of CAPE in contradistinction to other PLGA NPs.
Results and discussion:
(4) The author should add the data on the release behavior of CAPE from NPs.
Author Response
Response to the comments of Reviewer 2:
Introduction:
(1) The authors described the meaning of the improvement of the solubility of CAPE and the encapsulation of CAPE in PEG-PLGA.
RE: Raw CAPE has poor Solubility in most of solvents therefore we claim that PEG-PLGA is an excellent platform for encapsulation of CAPE with enhanced pharmacological activates in regards to wound healing.
(2) The solubility of CAPE should be described in specific figures.
RE: The reviewer comment is greatly appreciated. A solubility study wasn’t needed in the current work. PEG-PLGA efficiently encapsulated CAPE and showed appreciable physical and pharmacological properties.
(3) Why did the author choose PEG-PLGA? The author added a description of the properties of CAPE in contradistinction to other PLGA NPs.
RE: PEG-PLGA advantages have been described in more details in introduction
Results and discussion:
(4) The author should add the data on the release behavior of CAPE from NPs.
RE: We appreciate the constructive comment. Therefore, an in-vitro release study of the prepared CAPE-PEG-PLGA has been performed. Kindly check section 3.2

Round 2
Reviewer 1 Report
All the comments have been addressed in the revised manuscript.
Reviewer 2 Report
The authors rewrote their manuscript according to the reviewer's comments. Therefore, this manuscript would be acceptable as a regular article.